# Dynamic Phenotypic Switching and Group Behavior Help Non-Small Cell Lung Cancer Cells Evade Chemotherapy

**DOI:** 10.3390/biom12010008

**Published:** 2021-12-21

**Authors:** Arin Nam, Atish Mohanty, Supriyo Bhattacharya, Sourabh Kotnala, Srisairam Achuthan, Kishore Hari, Saumya Srivastava, Linlin Guo, Anusha Nathan, Rishov Chatterjee, Maneesh Jain, Mohd W. Nasser, Surinder Kumar Batra, Govindan Rangarajan, Erminia Massarelli, Herbert Levine, Mohit Kumar Jolly, Prakash Kulkarni, Ravi Salgia

**Affiliations:** 1Department of Medical Oncology and Therapeutics Research, City of Hope National Medical Center, Duarte, CA 91010, USA; anam@coh.org (A.N.); amohanty@coh.org (A.M.); sauravkotnala@gmail.com (S.K.); ssrivastava@coh.org (S.S.); lguo@coh.org (L.G.); anathan@coh.org (A.N.); emassarelli@coh.org (E.M.); pkulkarni@coh.org (P.K.); 2Integrative Genomics Core, Beckman Research Institute, City of Hope National Medical Center, Duarte, CA 91010, USA; sbhattach@coh.org; 3Center for Informatics, Division of Research Informatics, City of Hope National Medical Center, Duarte, CA 91010, USA; sachutan@coh.org (S.A.); rchatterjee@coh.org (R.C.); 4Center for BioSystems Science and Engineering, Indian Institute of Science, Bangalore 560012, India; kishorehari@iisc.ac.in (K.H.); mkjolly@iisc.ac.in (M.K.J.); 5Department of Biochemistry and Molecular Biology, College of Medicine, University of Nebraska Medical Center, Omaha, NE 68198, USA; mjain@unmc.edu (M.J.); wasim.nasser@unmc.edu (M.W.N.); sbatra@unmc.edu (S.K.B.); 6Department of Mathematics, Indian Institute of Science, Bangalore 560012, India; rangaraj@iisc.ac.in; 7Center for Neuroscience, Indian Institute of Science, Bangalore 560012, India; 8Department of Physics, Northeastern University, Boston, MA 02115, USA; h.levine@northeastern.edu; 9Department of Bioengineering, Northeastern University, Boston, MA 02115, USA; 10Department of Medical Oncology and Therapeutics Research, Arthur & Rosalie Kaplan Endowed Chair in Medical Oncology, 1500 E. Duarte Road, Duarte, CA 91010, USA

**Keywords:** chemoresistance, cisplatin, lung cancer, evolutionary game theory, group behavior, persister trait, phenotypic switching

## Abstract

Drug resistance, a major challenge in cancer therapy, is typically attributed to mutations and genetic heterogeneity. Emerging evidence suggests that dynamic cellular interactions and group behavior also contribute to drug resistance. However, the underlying mechanisms remain poorly understood. Here, we present a new mathematical approach with game theoretical underpinnings that we developed to model real-time growth data of non-small cell lung cancer (NSCLC) cells and discern patterns in response to treatment with cisplatin. We show that the cisplatin-sensitive and cisplatin-tolerant NSCLC cells, when co-cultured in the absence or presence of the drug, display dynamic group behavior strategies. Tolerant cells exhibit a ‘persister-like’ behavior and are attenuated by sensitive cells; they also appear to ‘educate’ sensitive cells to evade chemotherapy. Further, tolerant cells can switch phenotypes to become sensitive, especially at low cisplatin concentrations. Finally, switching treatment from continuous to an intermittent regimen can attenuate the emergence of tolerant cells, suggesting that intermittent chemotherapy may improve outcomes in lung cancer.

## 1. Introduction

Drug resistance in cancer is generally believed to arise stochastically through random genetic mutations and the subsequent expansion of mutant clones via Darwinian selection [1,2]. However, emerging evidence suggests that non-genetic and epigenetic mechanisms may also play a critical role [3,4,5], leading to enhanced adaptability and cooperation under stressful conditions [3,4,5]. Nonetheless, such mechanisms have not been fully explored and are rarely integrated into clinical trials or in precision oncology initiatives [6,7,8]. Furthermore, combining conventional therapies with treatment strategies based on cancer ecology could potentially delay or even prevent drug tolerance and eventually, drug resistance [2,8,9,10].

Several studies have reported the existence of drug-resistant and drug-tolerant (i.e., weakly or moderately resistant) clones in pre-treatment tumors [11], although the population of these clones is usually low in the presence of drug-sensitive cells [12]. This raises questions as to how drug-sensitive and resistant/tolerant clones in a tumor influence each other’s fitness (growth), and whether cooperation and competition (group behavior) between the sensitive and tolerant cells influence the response to drug therapy. Thus, discerning group behavior by monitoring interactions between drug-tolerant and -sensitive cells in real time in the absence or presence of the drug is a powerful tool to elucidate the role of group behavior [13,14].

Here, we have used human non-small cell lung cancer (NSCLC) cells that are sensitive or tolerant to cisplatin, one of the most commonly used chemotherapeutics, to understand the role of group behavior in the emergence of drug-tolerant clones, and eventually resistant clones. Ideally, these experiments would have been performed using cell lines that have a common evolutionary origin, e.g., from the same patient. However, sensitive and tolerant cells from the same patient were not readily available and our efforts in generating isogenic drug-tolerant cell lines by exposing drug-sensitive NSCLC cells to cisplatin was unsuccessful. Alternatively, we used established sensitive and tolerant cell lines to perform these experiments (see discussion for an explanation of the caveats). The cells expressing red fluorescent protein (RFP) or green fluorescent protein (GFP) were mixed (heterotypic culture) and cultured in different ratios. The proliferation of the two cell types was followed in real time and compared with the same cells grown alone (monotypic culture). The cell counts were used to determine the dynamic behavior of the population, and results were correlated with the cell-autonomous and non-cell-autonomous fitness (growth rate) effects [5]. Since cell-autonomous fitness effects are defined as those inherent to the cell, the growth rates from monotypic cultures provided the necessary information for determining these effects [4]. In contrast, non-cell-autonomous effects are those that allow fitness to depend on a cell’s microenvironment including the frequency of other cellular phenotypes as well as diffusible factors in the media [15].

Since standard models based on evolutionary game theory proved inadequate to analyze the data, we developed a new approach, Phenotypic Switch Model with Stress Response (PSMSR), that incorporates concepts from chemical reaction kinetics and the cooperative behavior of drug-tolerant phenotypes in the community. A distinguishing feature of the PSMSR model is that it considers the ability of cancer cells to switch phenotypes. Employing PSMSR, we showed that, quantitatively, the two cell populations, when co-cultured in the absence of cisplatin, display dynamic group behavior that can be interpreted using evolutionary game theory concepts as payoffs (benefit or loss of individual players associated with a set of game strategies such as competition or cooperation). Due to phenotypic switching by the cells, the game strategies were dynamically altered based on cell frequencies and stress level while maximizing group survival. However, in the presence of cisplatin, the group behavior (cooperation) was attenuated in favor of self-survival. Furthermore, we demonstrate that switching treatments from a continuous to an intermittent regimen can attenuate the emergence of tolerant cells, underscoring a potentially new treatment option that could benefit patients with NSCLC. These results can be further verified using sensitive and tolerant cells from the same patient or creating them from a common parental cell line, to make them more relevant in the broader cancer context. Nonetheless, initial success of the intermittent therapy in zebrafish, as shown in this work, suggests that our observations could be relevant in vivo.

## 2. Results

### 2.1. Cisplatin-Sensitive and Tolerant Cells Demonstrate Different Behaviors in Monotypic and Heterotypic Cultures

A schematic overview summarizing the experiments and the source of data collection used in developing the theoretical models are presented in Figure 1A–C. Fluorescently labeled cisplatin-sensitive H23 and cisplatin-tolerant H2009 NSCLC cells [16] were co-cultured and monitored in real time (Appendix A). Both the cell lines have frequently observed mutations of P53 and KRAS, in addition to STK11 and ATM specific to H23, or TERT and B2M mutation specific to H2009 cells (Cellosaurus cell line database). To discern differences in their behavior, the two cell line cultures were grown as monotypic (grown by themselves) or as heterotypic cultures (co-culture of sensitive and tolerant cells) in a 1:1 ratio. To determine the short-term effects of heterotypic culture, we incubated the cells for 12 h before the start of the experiment (Appendix A). But to determine the long-term effects, they were co-cultured for three weeks (without cisplatin) before the start of the experiment (Figure 2A, schematic).

At a 1:1 ratio, there was no significant difference in the fold-change (ratio of the cell population at any time point relative to t = 0 h) of the sensitive cell counts between the monotypic (grown by themselves) and heterotypic cultures (mixed with tolerant cells for 12 h or three weeks prior to counting), and they were equally sensitive to 5 µM cisplatin in all three conditions (Figure 2A, left panel bar graph in red). These monotypic and heterotypic culture experiments were also performed using tolerant cells, and no significant change in cell count or drug tolerance was observed for the cells mixed for only 12 h prior to the experiment (Figure 2A, right panel bar graph in green, bars labeled “Alone” and “Mix before”). However, when tolerant cells were co-cultured with sensitive cells for three weeks prior to the experiment, they showed a marked reduction in cell proliferation (Figure 2A green bar graph, dark green bar labeled “3-wk co-culture”). Moreover, when cisplatin was added to this 3-week co-cultured cells, the tolerant cells showed a smaller reduction in cell growth compared with those cultured separately or mixed 12 h before the experiment began (Figure 2A, right panel bar graph in green), and their proliferation was significantly attenuated by long coexistence with the sensitive cells, suggesting that tolerant cells appear to exhibit a ‘persister-like’ trait (please see the “Discussion” section) [17,18].

We further explored the long term 3-week co-culture experiments by seeding the cells at increasing tolerant-to-sensitive ratios (1:1, 2:1, 4:1, 8:1), and recorded their growth every 2 h using the IncuCyte live cell imaging system (Appendix A). At a seeding ratio of 1:1, the sensitive cells showed an 8-fold increase in cell count within 96 h, and reached a plateau post 96 h, whereas the tolerant cells exhibited a 4-fold increase in cell count within 72 h, followed by a drop and plateau for the rest of the time (Figure 2B). At the end of the experiment (144 h), the sensitive cell growth was 2.5-fold more than the tolerant cells for 1:1 ratio, compared with 1.5, 1.4 and 1.2-fold for the 2:1, 4:1 and 8:1 ratios, respectively (Figure 2C). These data reveal that the tolerant cell proliferation was suppressed in presence of sensitive cells but could be rescued by increasing the fraction of tolerant cells in the co-cultures.

Next, we determined the proliferation profile for all the seeding ratios in the presence of cisplatin. Cisplatin had a cytostatic effect on the sensitive cells, and the fold change in the cell count remained approximately 1 for all the ratios (Figure 2D, purple bar graph). However, the tolerant cell proliferation was approximately 1.4, 2.09, 1.95, and 2.04-fold at the tolerant:sensitive seeding ratios of 1:1, 2:1, 4:1 and 8:1, respectively. Therefore, the administration of cisplatin rescued the tolerant cell growth from the suppressive effect of the sensitive cells by selectively curtailing the sensitive cell growth. In fact, the tolerant cells proliferated better in the presence of cisplatin and increasing the seeding ratio of tolerant cells in the population also favored their growth (Figure 2D).

The sensitive cells have a suppressive effect on the tolerant cells in a frequency-dependent manner was also evident by analyzing the change in tolerant to sensitive fraction at the end of 144 h in the untreated conditions (0.2, 1.4, 6.3, 17.4 at T:S seeding ratios of 1:1, 2:1, 4:1 and 8:1, respectively, Figure 2E, black bars). In addition, the tolerant to sensitive fractions in the presence of cisplatin were 0.8, 4.9, 16.9 and 38.8 for the same seeding ratios (Figure 2E, orange bars). Thus, there was an increase in the tolerant to sensitive cell fraction in the population treated with cisplatin, suggesting a reduction in the sensitive cell population as well as better fitness of the tolerant cells in the presence of cisplatin.

### 2.2. Sensitive Cells Suppress Growth of Tolerant Cells in the Absence of Drug

To discern the effect of short-term association between the sensitive and the tolerant cells on their group behavior, we repeated the above experiments by incubating the co-culture for only 12 h instead of three weeks prior to starting the experiments (Appendix A). Here, at 1:1 tolerant:sensitive seeding ratio, the fold change in the tolerant cell population at the end of 144 h was 8-fold (Appendix A), whereas after three weeks of co-culture, we observed a 4-fold change in the growth (Figure 2C and Appendix A). Also, compared with the three weeks co-culture, where the tolerant cell fold changes were significantly lower (*p* < 0.0001) than that of the sensitive cells for all seeding ratios except 8:1, the 12-h co-culture showed less difference in fold changes between the two cell types (*p* < 0.001) at seeding ratios 2:1 and 4:1, and insignificant at 8:1 (Appendix A). The tolerant-to-sensitive fraction in the population at 144 h was also higher than in the three weeks co-culture experiments for seeding ratios 1:1 and 2:1 (1.2 and 3.3 vs. 0.2 and 1.4 in the three weeks co-culture, Appendix A). Taken together, these data show that the short-term association between the two cell types did not suppress the tolerant cell population as efficiently as the long-term association. However, in the presence of cisplatin, the tolerant cell proliferation was significantly higher compared with the sensitive cells (*p* < 0.0001, Appendix A). Next, we compared the change in the tolerant cells to the sensitive cell fraction and observed a similar trend as seen in the three weeks co-culture for all seeding ratios. The increase in growth of the tolerant cells was also supported by the increase in their fraction of the population in the absence of cisplatin, and further, that the presence of cisplatin supported their growth. (Appendix A).

The growth trends indicated a competition between the two cell types which was enhanced by long-term association. In contrast, the behavior of the tolerant cells suggested mutual cooperation to improve survival, which was favored by the higher seeding ratios. To explore the frequency-dependent competition of the sensitive cells towards the tolerant cells, we expanded the seeding ratios to increased proportions of sensitive cells in the population (i.e., sensitive to tolerant ratios of 1:1 to 8:1) (Appendix A).

Consistent with the previous experiments, in the absence of cisplatin, the sensitive cells suppressed the growth of tolerant cells (Appendix A). Again, in the presence of cisplatin, the tolerant cell growth was dominated in the population by approximately 1.9, 2.2, 2.7 or 3.8-fold over sensitive cells for the sensitive to tolerant ratios of 1:1, 2:1, 4:1 or 8:1, respectively (Appendix A). The increase in growth of the tolerant cells was also supported by the presence of cisplatin (Appendix A). Together, these experiments indicated that in a heterogeneous population, dynamic competition and cooperation exist between the sensitive and the tolerant cells, and the sensitive cells dominate over the tolerant cells. In contrast, the presence of cisplatin favors the survival and proliferation of the tolerant cells by inducing cell death among the sensitive cells.

### 2.3. Sensitive Cells Secrete a Factor(s) that Retards the Growth of Tolerant Cells

To discern whether a physical interaction between the two cell types is necessary or the sensitive cells secrete an ‘inhibitory factor’ to attenuate tolerant cell growth, the cells were grown in conditioned medium from sensitive or tolerant cell monocultures (Figure 2F, schematic). As seen in the right graph in Figure 2G, conditioned medium from the sensitive cells impeded proliferation of tolerant cells by ~4.5-fold. In contrast, the conditioned medium from the tolerant cells had no significant effect on the growth of the sensitive cells (Figure 2G, left graph), alluding to the presence of one or more inhibitory factors in the conditioned medium from the sensitive cells. Furthermore, at a lower seeding density (Figure 2H, schematic), the conditioned medium had a greater inhibitory effect (approximately 6-fold) on the tolerant cells and reduced in a dose-dependent fashion with an increase in tolerant cell seeding density (Figure 2I). Thus, the suppressive effect of the sensitive cell-conditioned medium was reduced with a greater number of tolerant cells (Figure 2I).

### 2.4. Intermittent Therapy Can Sustain a Population of Cisplatin-Sensitive Tumor Cells while Attenuating the Proliferation of Resistant Cells

Since the proliferation of the tolerant cells was remarkably impeded when co-cultured with sensitive cells for prolonged periods prior to cisplatin treatment, we asked if continuous or intermittent cisplatin treatments would differentially affect a mixed population of the two cell types. Toward this end, we mixed the two cell types at different sensitive-to-tolerant ratios and treated them as described in Appendix A. Within 10 days, we observed that the ratio of tolerant to sensitive cells increased by 50- to 100-fold for the initial seeding ratios of 1:1, 2:1 and 4:1 (sensitive:tolerant), respectively, under continuous treatment. On the other hand, the tolerant-to-sensitive (T:S) ratio for the intermittent therapy increased only 3- to 8-fold (Figure 3A–C), suggesting that sensitive cells were able to recover from the drug toxicity and proliferate.

We prolonged the intermittent therapy by splitting the cells growing in cisplatin-free media post cisplatin treatment into two sets: ‘Intermittent 1 cycle’ and ‘Intermittent 2 cycle’ as described in Appendix A. In ‘Intermittent 1’, we observed the (initially cisplatin exposed) cells cisplatin-free for 25 days, whereas in ‘Intermittent 2’, we treated the cells with one extra dose of cisplatin for four days, before observing them in cisplatin-free media for the rest of the duration. We then asked if the sensitive cells, once exposed to cisplatin, would outgrow the tolerant cells to recapitulate the data shown in Figure 2B and Appendix A.

We continued the culture for 24 days, to let the cells grow and once confluent, passaged one to five every 6–8 days. We observed that the tolerant vs. sensitive ratio fell to approximately 2, 1.6 and 0.4 for initial seeding densities of 1:1, 2:1 or 4:1, respectively, for the cells treated only once with cisplatin on Day 3 (Intermittent—1 cycle) (Figure 3D–F, black line). In contrast, the ratio of cells that received the second dose of cisplatin treatment (‘Intermittent—2 cycles’) and were allowed to recover in fresh media, did not show any decrease in the tolerant population and maintained a S:T ratio of 100/800 (Figure 3D–F, red line). To validate the in vitro observations in vivo, we injected zebrafish larvae with fluorescently tagged cells and treated them with cisplatin (see Appendix A for details). While continuous cisplatin treatment for five days resulted in a tumor with predominantly tolerant cells, intermittent treatment led to no significant change in the T:S ratio (Figure 3G), indicating that intermittent treatment can ensure a stable disease whereas the continuous therapy favors emergence of drug refractory disease. However, the behavior of sensitive and tolerant cells that share a common evolutionary origin and have co-evolved in the same host for a long period of time needs to be tested, something which we plan to address in future. In light of the fact that phenotypic switching is emerging as an important defense mechanism in cancer cells, our observations may have real implications in designing sustainable therapy.

### 2.5. Epigenetic Modulation Can Distinguish Drug Sensitivity, Tolerance and Resistance in Lung Cancer

To test the possibility that drug sensitivity can be regulated at the epigenetic level in a reversible way, as opposed to genetic mutations alone, we used two different epigenetic modulators namely, 5-azacytidine (5-AZA), a DNA methyltransferase inhibitor, and suberoylanilide hydroxamic acid (SAHA), a histone deacetylase inhibitor, and determined their effects on cisplatin resistance. While SAHA treatment did not enhance the effect of cisplatin on sensitive H23 cells (Figure 3H) or H1993 cells that are resistant to cisplatin (IC50 > 300 μM) (Figure 3J), it had a significant additive effect on the H2009 cells, suggesting that these cells can become sensitive through epigenetic intervention (Figure 3I). However, 5-AZA had no discernable effect (not shown), suggesting that epigenetic regulation of chromatin rather than specific cytosine residues in the DNA modulates cisplatin tolerance in the H2009 cells. Based on these criteria, H2009 qualify as cisplatin-tolerant (reversible) rather than resistant (irreversible) while H1993 may represent a truly resistant phenotype. Taken together, these observations suggest that tolerance to cisplatin can be reversed unless the tolerant cells acquire mutations making them irreversibly resistant.

### 2.6. Modeling Cancer Group Behavior Using Experimentally Derived Growth Curves

Typically, group behavior among cancer cell populations is studied using variants of the Lotka–Volterra (LV) model, where the inter-species competition and cooperation depend on the species frequencies [19,20,21]. We tested one such model (Li et al.) which has been successful in explaining the evolutionary dynamics in bacterial co-cultures [22]. The major difference of the Li et al. model to LV is the implementation of growth rates that depend on the species frequencies, leading to more complex dynamics than can be captured by the classical LV model. However, the Li et al. model did not quantitatively explain our experimental data (details in the Appendix A) warranting alternative and possibly more complex models. We think the models such as LV or Li et al. are not suitable for our system partly due to the fact that they treat each cell identity as immutable, therefore ignoring the plastic phenotypes of cancer cells due to phenotypic switching. To address these deficiencies, we have developed a new model (Phenotype Switch Model with Stress Response or PSMSR), incorporating the knowledge about our specific cellular system and the observed growth trends.

The key evidence that motivated the new model PSMSR are as follows:Sensitive cells suppress the proliferation of the tolerant cells by secreting diffusible factors (can be overcome by increasing the frequency of tolerant cells)The suppressive effect is only prominent after co-culture of the two cell types for three weeks, but not if the cells are mixed and monitored immediatelyCompetition by the sensitive cells is eliminated in the presence of cisplatinEpigenetic modifier SAHA can switch the tolerant cells to be drug-sensitive through non-genetic means, implying that these cells can switch their phenotypes in response to the environment.

Based on the above observations, the following are the key premises involved in PSMSR (Figure 4A):Sensitive cells generate one or more products that affect the proliferation of the tolerant cells (and possibly their own as well). We call this hypothetical product(s) ‘stress’ (we explain its significance in Appendix A).Since the cohabitation of the cells appear to change their phenotypes (e.g., stronger suppression of tolerant cells by the sensitive cells after three weeks of co-culture), and we do not have enough information to model this phenotypic change as function of the cohabitation conditions, every system (i.e., monotypic, heterotypic-12 h and heterotypic-3 weeks) must be treated as distinct with their own phenotypic parameters.Through mutual cooperation the tolerant cells can mitigate or neutralize the ‘stress’ generated by the sensitive cells in a frequency-dependent manner.Due to stochastic phenotypic switching (sensitive ⇌ tolerant) by the two cellular species, a state of equilibrium exists between the two phenotypes at any point of time, where the equilibrium constant depends on the stress. As stress increases in the system, the equilibrium shifts to the right to increase the fraction of the tolerant phenotype.To keep the model conceptually tractable, we make the simplifying assumption that in relation to the observed growth dynamics, there is no significant difference between the true tolerant phenotype and the one produced through phenotypic switching of the sensitive cells. Likewise, the true sensitive phenotype and the one from phenotypic switching are identical as well (see caveats mentioned in the discussion).

The model presented here reflects the following mechanisms: (i) cellular growth leads to stress accumulation, (ii) accumulated stress reduces growth, (iii) tolerant cells are efficient in neutralizing stress, (iv) stress accumulation triggers the switching of sensitive cells to the tolerant phenotype. Hence, the growth rates of the sensitive (*S*) and the tolerant cells (*T*) can be expressed as,
(1)$$\frac{dS}{\mathrm{dt}}=-K_{a}S+K_{b}T+K_{GS}S$$
(2)$$\frac{dT}{\mathrm{dt}}=-K_{b}T+K_{a}S+K_{GT}T$$

Here, *K_GS_* and *K_GT_* are the stress-dependent effective growth rates (incorporating both proliferation and cell death) of the sensitive and tolerant cells respectively. *K_a_* and *K_b_* are the rate of switching from sensitive to the tolerant phenotype and vice versa and K is the equilibrium constant of phenotypic switching (Equation (3)) [23]. We assume *K_GS_*, *K_GT_*, *K* to be linearly dependent on stress and *K_b_* to be fixed, although the exact functional forms that map these quantities to stress is less important, as long as a monotonic relationship is maintained. Notably, we also fit the PSMSR model assuming sigmoidal as opposed to linear relationships of the above rate parameters with stress (Appendix A), without significant worsening of fitting error (Appendix A). For details, see Appendix A.
(3)$$K=\frac{K_{a}}{K_{b}}$$
where *K* is the equilibrium constant for phenotypic switching.

Next, we assume that stress is predominantly generated by the fast-growing sensitive cells at a rate proportional to the cell population and neutralized by the tolerant cells. The resulting rate equation is given by:(4)$$\frac{dC_{Str}}{\mathrm{dt}}=K_{Str}S-K_{Str,d}T$$
where $C_{Str}$ is a hidden variable representing the stress level, and *K_Str_* and *K_Str,d_* are the rates of stress generation and removal, respectively.

### 2.7. PSMSR and Cisplatin Response

To model the effect of cisplatin, we added a cisplatin dose-dependent cellular death rate term to Equations (1) and (2) to obtain Equations (5) and (6). *AUC* stands for “area under the curve” (*AUC* = cisplatin concentration × time of exposure), which represents the memory effect of cisplatin exposure on the cellular growth (Figure 5A and Appendix A) [24,25,26]. *Sigmoid*(*AUC*) is the sigmoidal function (Equation (7)) that varies between 0 and 1, depending on the magnitude of *AUC*, which multiplied by the scale factor SCALE gives the cellular death rate. *AUC*_5_ and *AUC*_95_ represent the *AUC* values where 5% and 95% of the cisplatin death effect are achieved, respectively. The SCALE, *AUC*_5_ and *AUC*_95_ parameters are specific for the sensitive and the tolerant cell types.
(5)$$\frac{dS}{dt}=-K_{a}S+K_{b}T+K_{GS}S-{\mathrm{SCALE}}_{S}\times sigmoid\left(AUC\right)\times S$$
(6)$$\frac{dT}{dt}=-K_{b}T+K_{a}S+K_{Gt}T-{\mathrm{SCALE}}_{t}\times sigmoid\left(AUC\right)\times T$$
(7)$$sigmoid\left(AUC\right)={\left\{1+exp\left[ln19\left(1-2\frac{AUC-AUC_{5}}{AUC_{95}-AUC_{5}}\right)\right]\right\}}^{-1}$$

In total, the PSMSR includes 9 unknown parameters (16, when including cisplatin effect). We used roughly 2900 cell population data collected over several days of cellular growth under various conditions to fit (Figure 4B,C) and analyzed the model parameters. The fitting was performed using the global optimization method, Genetic Algorithm (GA) [27], as implemented in the package GA in R [28] (also see Appendix A). The suitability of the PSMSR model in describing the experimental observations was assessed by constructing the log likelihood profiles for each parameter as described in Appendix A.

### 2.8. PSMSR in Monotypic and Heterotypic Cultures

Our experimental observations indicated that the sensitive and the tolerant cells behave quite differently when cultured alone when compared with coculture (Appendix A). Therefore, by deconvoluting the growth trends using PSMSR, we examined whether a few or all parameters of the model are different between the monotypic and the heterotypic cultures. In general, several phenotypic parameters for the heterotypic cultures were different in magnitude compared with the values for the monotypic cultures (Appendix A). This indicates an influence of the cellular phenotypes on each other. The parameters that showed a consistent difference between the mono- and heterotypic cultures include *K*_0_ and *K_b_* (the parameters for phenotypic switching), *K_Gt_*_0_ (growth rate for the tolerant phenotype) and *K_s_* (rate of stress generation). One interesting observation is that the growth rate for the tolerant phenotype in monotypic and 3-week co-cultures is 7–10 times smaller than that of the sensitive phenotype (Appendix A), reminiscent of a persister-like trait seen in microbial systems (please see “Discussion”).

By incorporating the cellular growth dynamics under different seeding ratios, The PSMSR model has the potential to provide insights into the mechanism of phenotypic switching in response to a changing microenvironment (Figure 4D–I). Here, we have calculated the emergence of tolerant cell population from sensitive cells or vice versa in both monotypic (Figure 4D,E and Appendix A) and heterotypic cultures (Figure 4G,H and Appendix A). Of note, these cells are still experimentally detected as red or green, irrespective of their true phenotypes. Since in our model, stress is the driver for phenotypic switching, the switched phenotype cells only appear once stress builds up in the system over time (Figure 4I and Appendix A). In both monotypic and heterotypic cultures, the model predicts a rapid switch by the tolerant cells to the sensitive phenotype (up to 97%, Appendix A), while maintaining a low but steady tolerant population throughout the observations (Figure 4E and Appendix A). Thus, we have seen how the cancer cells use phenotypic switching to maintain the overall fitness of the community under different stress levels. To maintain steady growth, stress must be mitigated, where switching to the tolerant phenotype pays off, since the tolerant cells are capable of neutralizing stress. However, the fraction of tolerant phenotype in the population is predicted to be low overall (10%, Figure 4G and Appendix A). This may be due to a balance between the necessity for stress removal and the energy or other costs required to maintain the tolerant phenotype.

### 2.9. Effects of Phenotypic Switching and Stress Give Rise to Diverse Game-Theoretical Strategies in Mixed Cell Populations

Cancer cell behavior is widely studied using game theory-based models where the inter-species game strategies (competition and cooperation) are assumed to be constant throughout the growth regime. A familiar example of such a model is the competitive Lotka–Volterra equation, although more specialized models exist in the literature [29]. The phenotypic diversity available to cancer cells suggest that their game strategic landscape will be considerably complex, where the inter-species competition and cooperation are dynamically altered based on changing scenarios.

Therefore, we asked whether the PSMSR model can capture this complex strategic landscape. Due to the complexity of the PSMSR equations, it is not possible to analytically derive evolutionary payoffs (such as those given by the Lotka–Volterra equations [30]). Moreover, the payoffs are likely to vary over time, unlike in the classical game theoretical models, where they are assumed to be constant. Therefore, we followed a different approach, where we numerically fitted the Lotka–Volterra equation to the growth rates given by PSMSR over moving time windows (Appendix A). At small (12 h) and very large time windows (6 days), the fitting seemed to be worse, while at moderate time windows such as 4–5 days, the fitting appeared reasonable between the two models (Appendix A). By fitting the growth rates obtained from the PSMSR model to the competitive Lotka–Volterra equations, we determined the inter-species competition parameters as function of time, for different sensitive to tolerant seeding ratios (Figure 4J,K). Figure 4J shows that the tolerant cells are initially competitive towards the sensitive cells, but they become cooperative after 2–3 days of growth. This coincides with the accumulation of stress (Figure 4G) indicating that the microenvironment plays a major role in altering the game strategies of the cancer phenotypes. Notably, the effect of the sensitive cells towards the tolerant cells (Figure 4K) is significantly smaller in magnitude. Within the tumor, where the microenvironment is significantly more complex than our experimental setup, multiple agents such as the various cancer-associated macrophages and immune cells can dynamically alter the game strategies adopted by the tumor cells and steer resistance evolution [31]. In summary, the PSMSR model, combined with the payoff calculation scheme described above, demonstrates the diverse strategic landscape explored by the sensitive and tolerant cells under varying cell population and stress levels.

### 2.10. PSMSR Model Demonstrates the Effectiveness of the Intermittent Cisplatin Therapy

Using the PSMSR model, we simulated the continuous and intermittent therapy experiments, as explained before. We first calculated the model parameters by fitting to the experimental growth trends measured in presence of cisplatin. The best agreement was obtained when community cooperation such as phenotypic switching and stress removal were turned off (Figure 5B,C). This implies that high cisplatin levels trigger the tolerant cells to focus on their own survival, similar to other social communities where imminent danger promotes self-survival. Also, at high cisplatin levels, the phenotypic switching to sensitive is detrimental to the survival of the community. Together, these observations are indicative of the adaptability of the cancer cells for survival in adverse environments. In Figure 5D, the magnitudes of the SCALE parameter quantify the difference in cisplatin sensitivity between the sensitive and the tolerant phenotypes.

Using the PSMSR model, we simulated the cisplatin dose cycles as explained in Appendix A. Analogous to the experimental observations, the tolerant cell proportion increased with time in the continuous therapy, while it remained relatively small and increased at a slower rate during the intermittent cycles (Figure 5E,F). Also, the intermittent 2 cycles of cisplatin dosage created more tolerant cell population than the single cycle, in agreement with the experiments (Figure 3D–F). While the actual magnitudes of the tolerant cell population in the simulations are different than in the experiments, the qualitative behaviors agree. The quantitative difference between the predictions and the experiments could be partly attributed to the growth attenuation due to confluence that is not accounted for by the PSMSR model. Overall, these simulations show that the PSMSR model is able to qualitatively reproduce the drug-induced behavior of the cancer cell population.

## 3. Discussion

Several studies have applied evolutionary game theory to cancer [3,4,5,9,29,32,33,34,35] but as far as we are aware, the adaptive strategies NSCLC cancer cells adopt in response to environmental perturbations have not been investigated employing drug-naïve and drug-tolerant cells. We demonstrated that the growth dynamics of these cells can only be explained by invoking dynamic phenotypic switching upregulated by environmental stress. Consistent with this assumption, the present data with the epigenetic regulator together with our previous studies [16,36] and those from others [37,38], strongly support the possibility that the two cell types can stochastically switch their phenotypes via non-genetic mechanisms.

The present study also highlights the complex behavioral landscape of cancer cells, where the payoff strategies are dynamically evolving via phenotypic plasticity induced by environmental pressure (Figure 4J,K). These inner level traits are cell frequency dependent and can affect the carrying capacity of the population. The proposed PSMSR model can provide important information about dynamic payoff strategies of multiple cellular phenotypes in a time and frequency-dependent manner. In contrast, traditional evolutionary models (e.g., the competitive LV) hold those payoffs to be fixed but are otherwise useful in understanding broad game strategies of the system, due to their mathematical simplicity. Combining the PSMSR with a model analogous to LV can thus give additional insight into the complex group behavior of multiple cellular phenotypes including the role of the microenvironment, as was demonstrated in Figure 4J,K.

The mathematical modeling combined with the experimental observations demonstrated that accumulated stress (induced by cell growth and microenvironment), promotes phenotypic switching of sensitive cells to tolerant phenotypes. Thus, cells that switch phenotypes help to partly neutralize the stress and allow the fast-growing sensitive cells to proliferate, thereby sustaining the carrying capacity of the system. Thus, the resulting carrying capacity is a function of both stress and the level of tolerant phenotype in the system. Interestingly, the behavior of the tolerant cells appeared to be beneficial to the overall community. They helped the proliferation of the sensitive cells by removing stress, and they themselves adopt to a slower proliferation rate (when co-cultured for three weeks with the sensitive cells, the parameter *K_Gt_*_0_ in Table 1), so as to not compete for limited resources (altruism). This altruistic behavior is even more beneficial in the crowded environment of a tumor, where nutrients and oxygen could run low. The tolerant cells elucidated the evolutionary strategy of bet-hedging where they display low evolutionary fitness under normal conditions, but high fitness under stressful conditions, such as in the presence of cisplatin [39]. The intermittent therapy simulations show that the PSMSR model reproduces this behavior of the tolerant cell population (Figure 5E,F). Comparing the growth data with and without cisplatin in conjunction with the PSMSR model, we also find that the phenotypic switching (and the consequent altruism) may be turned off at high stress, when cisplatin is administered. Therefore, it appears that the altruistic stress removal benefit by the tolerant cells could be effective under normal conditions in the tumor, where a small tolerant cell population benefits the drug-sensitive cells to sustain proliferation. However, under the high stress of chemotherapy, such stress removal mechanisms may be insufficient to sustain the sensitive cell viability. It is therefore prudent to turn off phenotypic switching under such situations and allow the sensitive cells to become extinct and the tolerant cells to proliferate. While bet-hedging strategies by drug-tolerant phenotypes are well discussed in the literature [39], altruism by such phenotypes has hitherto been unexplored. There is overwhelming evidence that such tolerant persister phenotypes exist in the tumor in small proportions, even in non-drug-resistant disease. However, it is not clear whether they have a specific ecological role other than adding to the tumor heterogeneity, although recent evidence indicates that persisters can facilitate escape from drug induced toxicity by reversibly switching to slow cycling phenotypes [18].

Taken together, the behavior of the tolerant cells provides novel insights into the phenotypic traits in cancer that emerge due to survival pressure as well as the cost-benefit basis of such evolution. Of note, when co-cultured for three weeks prior to starting the experiment, both the sensitive and tolerant cell types had equal opportunity to switch their strategy; either they could have reduced or increased their proliferation rate to compete, but they followed an unexpected path where the sensitive cells remained unaffected, and the tolerant cells reduced their proliferation rate. This strategy could be helpful because most of the genotoxic drugs used for chemotherapy target actively dividing cells. This behavior of tolerant cells resembles phenotypic switching behavior reminiscent of persisters that are well known in microbial systems [40] and have more recently been recognized in cancer [11,22,39,41,42].

Admittedly, a potential caveat of the current study is that the sensitive and the tolerant cells were harvested from different patients and hence do not share a common genetic or ecological origin. Therefore, an argument can be made that at least some of their behavior in a co-culture environment could be attributed to the cell line specific differences. This caveat aside, the current work depicts the behavioral complexity of cancer cells under different ecological conditions and potential emergence of novel evolutionary strategies that cannot be forecast without the aid of mathematical modeling. Most importantly, the sensitive and the tolerant cells exhibit cooperation, which is unexpected given these cells are from unrelated origins. When cells or species from different ecological environments are brought together, the most expected behavior is either competition (e.g., invasive species) or neutrality (if there is no resource limitation). In contrast, the observation of cooperation by the tolerant cells informs us about their evolutionary behavior towards similar drug sensitive cells in the patient of origin. Another notable fact is that the behavior of the sensitive cells could be modulated by three weeks of co-culture with the tolerant cells. This also informs us about the co-evolution of drug-sensitive and drug-tolerant phenotypes and cannot be explained by the cell line specific differences. Nevertheless, we believe, further experiments need to be performed to fully understand the extent of group behavior by using cells from a common origin.

From a translational perspective, the present study also suggests that intermittent rather than continuous chemotherapy may result in better outcomes in lung cancer. Although it may not cure the patient of the disease, it could potentially result in a stable disease that can be managed while sparing the patient the undesirable effects of excessive chemotherapy. The fact that intermittent therapy has shown promise in other solid tumors [7,9] should serve as a motivation to try it in lung cancer. Further in vitro and animal experiments using sensitive and tolerant cells from the same patient will establish the relevance of our observations in the broader context of cancer therapy.

## 4. Materials and Methods

The details of the cell lines, antibodies and reagents used, the protocol used for live cell imaging, the ratios in which the heterotypic cultures were grown and observed in real time along with the details of the zebrafish microinjection, drug treatment, and animal handling, are provided in full detail in the Appendix A. The cartoon images were made using the BioRender software.

## Figures and Tables

**Figure 1 biomolecules-12-00008-f001:**
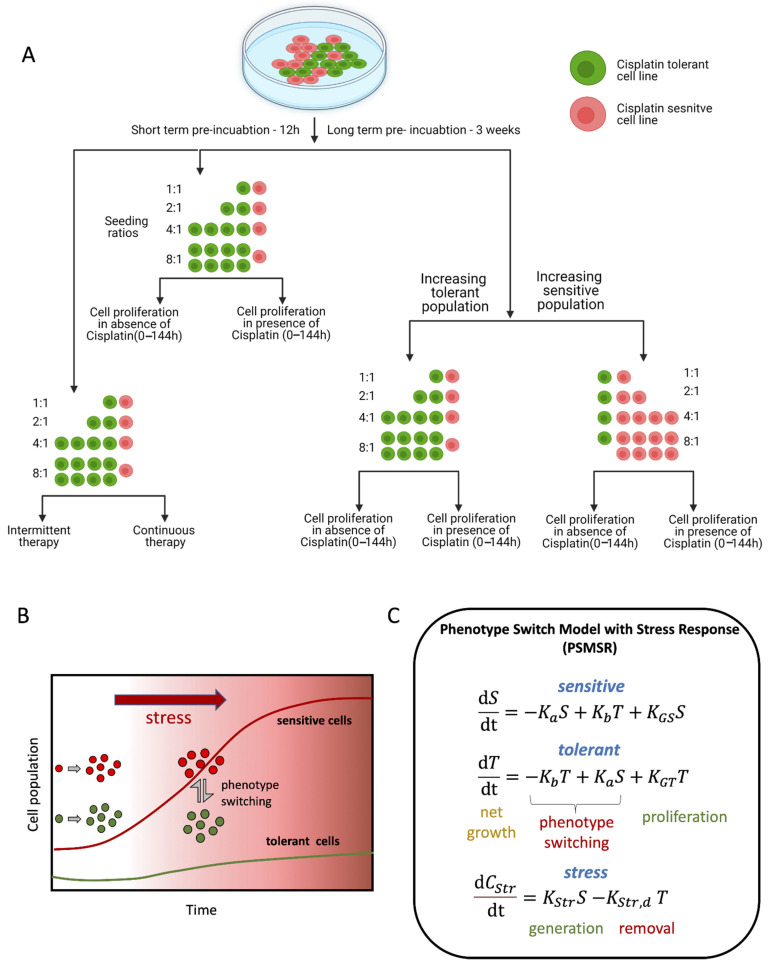
**Schematic summarizing the experiment and the source of data used in developing the theoretical cell growth models.** (**A**) The schematic representation of the different incubation duration, ratios, and treatments used for generating the data to develop the mathematical model. (**B**) schematic describing the principles on which the mathematical model PSMSR was developed; (**C**) panel representing the functional form of PSMSR (please see the main text for further details).

**Figure 2 biomolecules-12-00008-f002:**
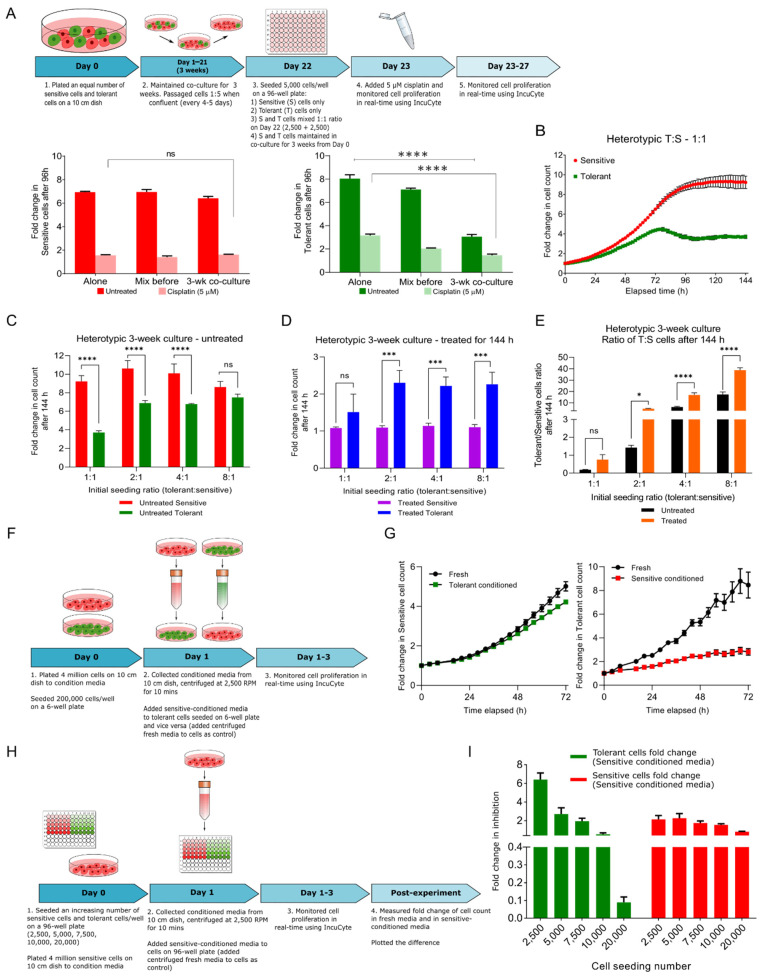
**Behavior of cisplatin-sensitive (S) and tolerant (T) NSCLC cells in 2D co-culture.** (**A**) Schematic representation of the experimental design of co-culturing S and T cells in a ratio of 1:1 and collection of data points. Proliferation of sensitive (red) and tolerant (green) cells under different culture conditions in the absence or presence of cisplatin. Two-way ANOVA test (multiple comparison) showing statistical significance **** *p* < 0.0001. (**B**) Sensitive and tolerant cells were plated in increasing T:S ratios and cultured for three weeks. Proliferation rate of sensitive cells (red) and tolerant cells (green) in heterotypic culture over the course of 144 h. (**C**) Fold change in cell count of sensitive cells (red) and tolerant cells (green) in heterotypic culture was measured after 144 h for ratios 1:1, 2:1, 4:1 and 8:1. Two-way ANOVA was used for calculating statistical significance **** *p* < 0.0001, ns—not significant. (**D**) Fold change in cell count of sensitive cells (purple) and tolerant cells (blue) in heterotypic culture was measured after 144 h in presence of cisplatin for ratios 1:1, 2:1, 4:1 and 8:1. Two-way ANOVA was used for calculating statistical significance *** *p* < 0.0001, ns—not significant. (**E**) Change in tolerant/sensitive cells ratio with (orange) and without (black) 5 μM cisplatin over the course of 144 h was measured. Statistical significance * *p* ≤ 0.05, **** *p* < 0.0001, ns—not significant. (**F**) Schematic representation of the conditioned medium experiment. (**G**) The left line graph representing the effect of tolerant cell conditioned medium on sensitive cells, and the right line graph representing the inhibitory effect of tolerant cell conditioned medium on sensitive cells growth. (**H**) Schematic representation of conditioned medium experiment to correlate the stoichiometry between cell number and inhibitory effect secreted by sensitive cells. (**I**) The bar graph representing the inhibitory effect of condition medium on different cell number of tolerant or sensitive cells. Statistical significance information can be found in Appendix A.

**Figure 3 biomolecules-12-00008-f003:**
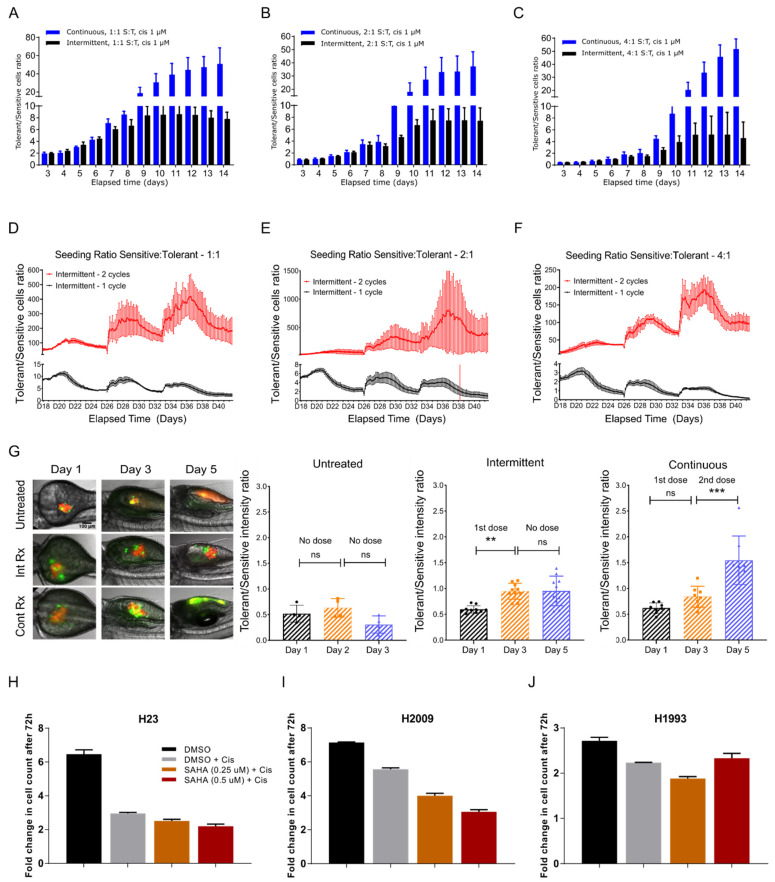
**Tolerant cells reversibly switch their phenotype to become sensitive with intermittent therapy.** (**A**–**C**) Bar graph showing the ratio of tolerant versus sensitive cell population over a period of 10 days. The cell ratio for the “Continuous” group wherein the cells were continuously treated with cisplatin is shown in blue and the ratio for the “Intermittent” group wherein the cells were treated with cisplatin for two days and released in fresh medium (intermittent) is shown in black. (**D**–**F**) Media from “Intermittent—2 cycles” group was removed after four days of cisplatin treatment and replaced with fresh medium and the cells were allowed to grow until confluent. These cells were monitored in real-time to determine the ratio of tolerant versus sensitive over the course of 25 days. Similarly, the cells that only received cisplatin once (“Intermittent—1 cycle”) throughout the experiment were also followed for 25 days. (**G**) Sensitive (S, red fluorescence) and tolerant (T, green fluorescence) cells were mixed at S:T ratio of 4:1 and microinjected into the perivitelline space of zebrafish larvae 48-h post fertilization (hpf). Twenty-four hours after microinjection, larvae were randomly divided into three groups: Group 1 received no drug treatment (Untreated), Group 2 received cisplatin 20 µM for three days and released with no drug for two days (Intermittent), and Group 3 received cisplatin 20 µM continuously for five days (Continuous). Ratio of tolerant versus sensitive cells was determined by measuring fluorescence intensity. One way ANOVA was used for calculating statistical significance ** *p* < 0.05, *** *p* < 0.001, ns—not significant (**H**–**J**) Effect of suberoylanilide hydroxamic acid (SAHA) on cisplatin-sensitive (H23), -tolerant (H2009), and -resistant (H1993) cells, demonstrating that SAHA can reverse the phenotype of H2009 from tolerant state to sensitive state. Statistical significance information can be found in Appendix A.

**Figure 4 biomolecules-12-00008-f004:**
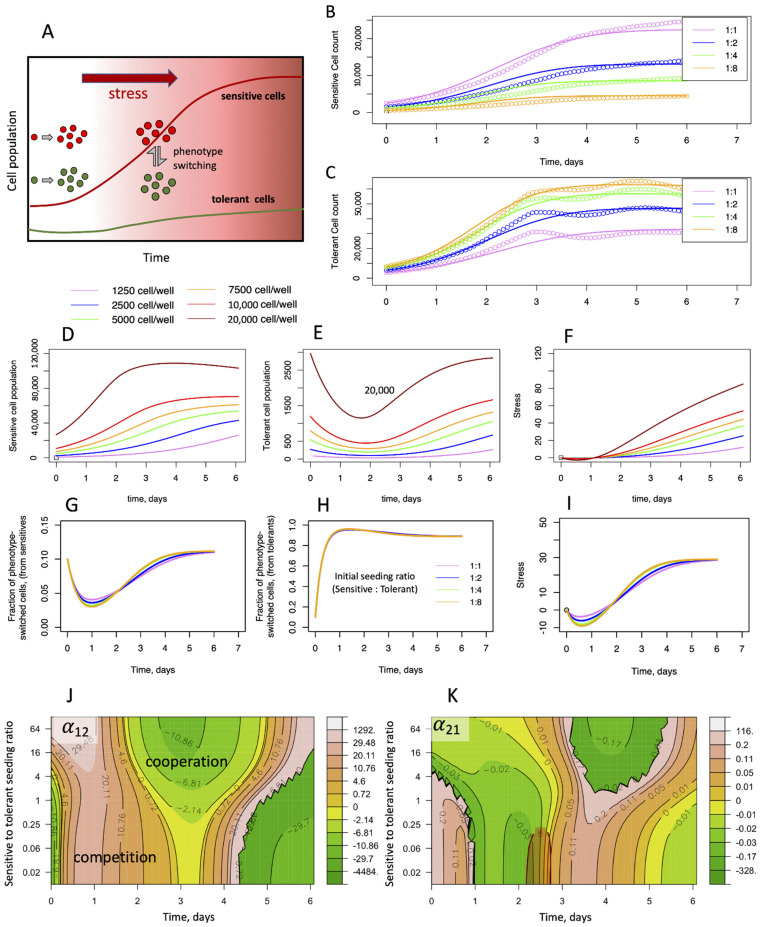
**Cooperativity and stress response as described by the PSMSR model.** (**A**) Schematic describing the PSMSR model; initially, the sensitive and the tolerant cells proliferate independently; as stress builds up, sensitive cells switch their phenotype to tolerant cells and vice versa; tolerant cells remove stress and maintain a small population, while enabling the sensitive cells to proliferate. (**B**,**C**) Fitting of the phenotype-switch model to the cellular growth curves of sensitive and tolerant cell populations, where the cells were mixed at different proportions and counting was started immediately; the colors represent the growth curves from different initial seeding proportions, as indicated in the legend (sensitive to tolerant cell seeding ratios); (**D**–**F**) predicted evolution of phenotypic switching and stress in monotypic cultures; (**D**,**E**) populations of sensitive and (switched) tolerant phenotypes with time, when seeded with sensitive cells only; (**F**) stress as function of time; (**G**–**I**) predicted evolution of switched phenotypes and stress in heterotypic culture experiments, where cell growth was monitored immediately after mixing; (**G**) fraction of sensitive cells that have switched to the tolerant phenotype, as function of time; (**H**) fraction of tolerant cells that have switched to the sensitive phenotype, as function of time; (**I**) stress with time; colors are according to the initial seeding ratio of sensitive to tolerant cells as shown in the legend; the total cell population in each case was close to 5000; (**J**,**K**) evolving game strategy landscape of cellular population due to stress and phenotypic switching; the heatmaps of time varying payoff values representative of inter-species competition/cooperation are shown as function of the sensitive-to-tolerant seeding ratio; payoff values are derived by fitting the PSMSR model to the competitive Lotka–Volterra equations; orange areas in the maps represent competitive behavior, green areas represent cooperative behavior; (**J**) α_12_ representing the effect of tolerant cells towards the sensitive cells; (**K**) α_21_ representing the effect of sensitive cells towards the tolerant cells.

**Figure 5 biomolecules-12-00008-f005:**
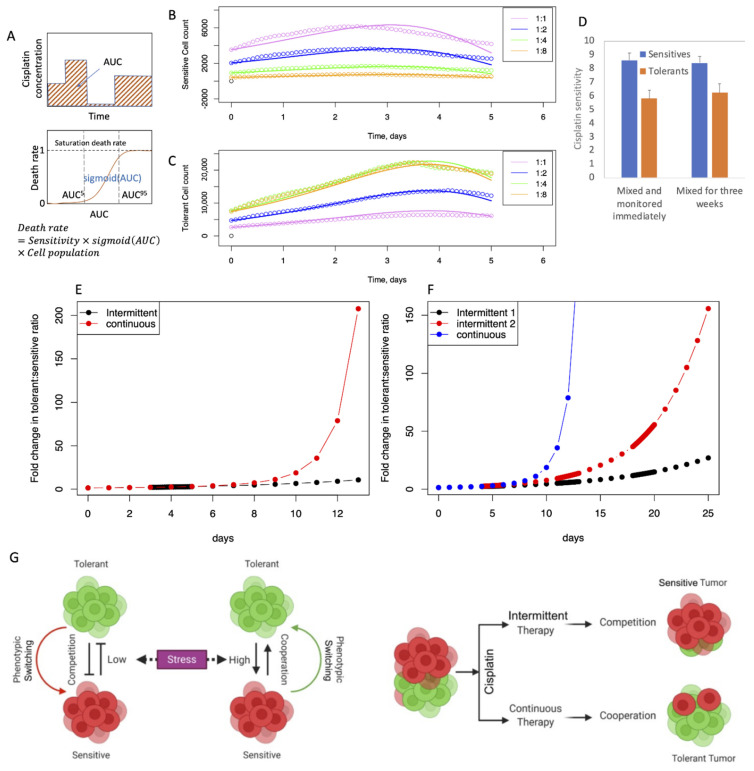
**Mathematical model for cisplatin resistance.** (**A**) Schematic demonstration of *AUC* and cellular death rate as function of *AUC*; (**B**,**C**) fitting of the experimental growth data where the cells were co-cultured for three weeks; (**B**): sensitive cells; (**C**): tolerant cells; circles and lines represent the experimental and fitted trends respectively; (**D**) SCALE parameter as measure of cisplatin sensitivity for the sensitive and the tolerant cells; the error bars represent 95% confidence limits (**E**,**F**) simulation of intermittent and continuous cisplatin treatment according to the protocols described in Figure 3; the initial sensitive to tolerant cell ratio was set to 4:1 with a total cell population of 50,000. (**G**) An illustrative model depicting the presence (and absence) of group behavior among sensitive and tolerant cells under varying conditions of stress and effects of continuous versus intermittent therapy.

**Table 1 biomolecules-12-00008-t001:** Model parameters and parameter search ranges for PSMSR, including the 95% confidence limits.

Condition	*K*	*K_b_*	*K_Gs_* _0_	*K_Gt_* _0_	*K_str_*	*K_str,d_*	*a*	*b*	*g*	*S_drug_*	*AUC_s_* ^5^	*AUC_s_* ^95^	*SCALE_s_*	*AUC_t_* ^5^	*AUC_t_* ^95^	*SCALE_t_*
Heterotypic	0.049 ± 0.0003	3.57 ± 0.02	0.713 ± 0.001	0.687 ± 0.004	7.14 × 10^−4^ ± 3 × 10^−6^	5.64 × 10^−3^ ± 4 × 10^−5^	0.046 ± 0.0004	0.038 ± 0.0002	0.018 ± 0.0006							
Heterotypic, 3 weeks	0.052 ± 0.0007	2.6 ± 0.05	0.708 ± 0.002	0.189 ± 0.016	6.74 × 10^−4^ ± 8 × 10^−6^	5.52 × 10^−3^ ± 1 × 10^−4^	0.05 ± 0.001	0.038 ± 0.0006	0.02 ± 0.0006							
Heterotypic, cisplatin 5 μM	0.033 ± 0.0017	0	1.033 ± 0.046	0.976 ± 0.054	5.9 × 10^−4^ ± 2.8 × 10^−5^	0	NA	0.04 ± 0.003	0.034 ± 0.003	108 ± 14.5	239 ± 12	1153 ± 58	8.61 ± 0.5	233 ± 17	1201 ± 49	5.79 ± 0.6
Heterotypic, 3 weeks, cisplatin 5 μM	0.033 ± 0.0015	0	0.966 ± 0.05	0.967 ± 0.05	5.8 × 10^−4^ ± 3.3 × 10^−5^	0	NA	0.039 ± 0.002	0.036 ± 0.003	96.8 ± 9.95	271 ± 14	1098 ± 44	8.91 ± 0.5	250 ± 14	1098 ± 47	6.89 ± 0.7
Parameter search range	0–0.1	0–5	0–2	0–2	0–0.001	0–0.02	0–0.1	0–0.1	0–0.1	1–500	1–500	10–1500	0.1–20	1–500	10–1500	0.1–20

## Data Availability

The codes used in our data analysis have been deposited in a public repository on Github. The link to the repository is: https://github.com/ID6ERS/Cancer_Resistance_Math_Modeling.

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
