# Peer review of "Dynamic Phenotypic Switching and Group Behavior Help Non-Small Cell Lung Cancer Cells Evade Chemotherapy"

_biomolecules, 2021, doi:10.3390/biom12010008_

Round 1

Reviewer 1 Report

In the research paper entitled “Dynamic phenotypic switching and group behavior help non-2 small cell lung cancer cells evade chemotherapy”, the authors tried to show phenotypic switching of drug resistance and proliferative capabilities between cisplatin tolerant and sensitive cell lines. As the authors already mentioned, the main drawback of the paper is not using the cells from the same patient. There are a few more drawbacks that authors should consider

  1. No cell line match
  2. The authors showed that sensitive cells secrete some factor that blocks proliferation. Authors are finding it because the cells are from different origins genotypically. If the clones are of the same genotypic origin, we may not even find this variation.
  3. Paper was written, including multiple cell dilutions and incubation which confuses the reader. Please either rewrite the paper or omit a few combinations wherever necessary without losing the story.

If the authors cannot generate a drug-resistant clone, they should approach labs which already established them, to procure the cell lines. Few suggestions for cell lines (PMID: 32446175, 32637578, 28746345). If authors can get the A549-CR cell line which is widely used in many labs and could replicate similar results, then it gives value to the work.

Author Response

Reviewer #1 Comments and Suggestions for Authors

In the research paper entitled “Dynamic phenotypic switching and group behavior help non-2 small cell lung cancer cells evade chemotherapy”, the authors tried to show phenotypic switching of drug resistance and proliferative capabilities between cisplatin tolerant and sensitive cell lines. As the authors already mentioned, the main drawback of the paper is not using the cells from the same patient. There are a few more drawbacks that authors should consider

  1. No cell line match

Author’s response: Unfortunately, we are unclear what the reviewer means by “no cell line match”. If the reviewer is referring isogenic cell lines, please see our response below.

  1. The authors showed that sensitive cells secrete some factor that blocks proliferation. Authors are finding it because the cells are from different origins genotypically. If the clones are of the same genotypic origin, we may not even find this variation.

Author’s response: Inasmuch as we appreciate the reviewer’s comment (and keeping in mind the pitfalls of the current study that were already explained in the manuscript), one could also argue that even If the clones are of the same genotypic origin, one could find this variation in growth rate. For example, please see Kaznacheev et al. Nat Ecol Evol 2019 (PMID: 30778184). Here the authors report evolutionary game in mixed cocultures of sensitive and resistant NSCLC cells obtained from the same parental cell line. In this work, the resistant cells showed significantly different growth rate in mixed culture compared to monoculture, although the sensitive and the tolerant cells were from the same parental cell line. Several recent reports in the literature have reported distinct clones within the tumor that promote or suppress growth of other clones, even when they share the same evolutionary lineage (e.g., Clearly et al., PMID: 24695311; Archetti et al., 25624490). Lastly if this was only a competition between cells of different genetic origin, one would expect to see this competition offered by both the sensitive as well the tolerant cells. However, the tolerant cells in our study do not compete with the sensitive cells, it’s only the sensitive cells that compete with the tolerant cells.  

  1. Paper was written, including multiple cell dilutions and incubation which confuses the reader. Please either rewrite the paper or omit a few combinations wherever necessary without losing the story.

Author’s response: We apologize for the confusion. We have now revised the text to add clarity.

  1. If the authors cannot generate a drug-resistant clone, they should approach labs which already established them, to procure the cell lines. Few suggestions for cell lines (PMID: 32446175, 32637578, 28746345). If authors can get the A549-CR cell line which is widely used in many labs and could replicate similar results, then it gives value to the work.

Author’s response: We thank the reviewer for these suggestions. However, the goal of the current work is to analyze the group behavior of tolerant cancer cells whose drug tolerance is reversible, rather than permanently resistant cancer cells. As such, the cell lines that we have used satisfied our criteria (also please see Mohanty et al, iScience 2020, PMID: 32947124). That being said, it is certainly our aim to expand these observations in future with other resistant cell lines.

We would like to mention that we purposefully chose to use naturally occurring drug sensitive and ‘tolerant’ (clones that are weakly or moderately resistant) cell lines that were developed from NSCLC cells isolated from a patient who was drug naïve and from a patient who was sensitive but developed tolerance subsequently, respectively. Furthermore, our choice of naturally occurring sensitive and tolerant cell lines reflects the natural circumstances (in the tumor microenvironment) under which the cells developed tolerance rather than a more artificial environment when the cells are grown in a petri plate. The tolerant cells exhibit the hallmark characteristics of phenotypically plastic persisters, which have been recently reported in the literature. We respect the caveat that these cells did not originate from the same parental lineage. However, it is unlikely that the major behavioral features of these cell types in presence of one another can be entirely attributed to their genetic differences. For example, the retarded growth of persisters in presence of sensitive cells is a well-documented observation in the literature (PMID: 34352380, 30297799). Also, the cooperation of the tolerant cells with the sensitive cells is a novel observation that cannot be explained by invoking competition between unrelated cell lines.

Reviewer 2 Report

Nam et al.

Dynamic phenotypic switching and group behavior help non small cell lung cancer cells evade chemotherapy

The author describe a model of cultured lung cancer cells with different genetic background and analyzed their behavior and interactions under the exposition with chemotherapeutic drugs. They used mathematical modelling to predict the cellular behavior under these conditions.

They concluded that coculture of sensitive and tolerant cells revealed that there is a dynamic competition and collaboration among these cell groups. Without cisplatin the sensitive cells dominate the tolerant cells but under cisplatin treatment the tolerant cells evade treatment with prolonged survival and proliferation whereas the sensitive cells undergo cell death. Further experiments were performed with tolerant and sensitive cells alone but with conditioned medium of one and another. This revealed an inhibitory effect on tolerant cells with conditioned medium of sensitive cells but not for the opposite experiment. Further experiments were performed to compare different drug exposure regimens and monitor sensitive and tolerant cells in an in vivo experiment with zebra fish larvae. Continous exposure of the drug showed a significant decrease in tolerant cells whereas intermittent drug exposure did not. They also tested the modulation of epigenetic changes by applying 5-AZA and SAHA showing that epigentic modulation is able to discern drug tolerance vs. true drug resistance.

The presented data are very interesting and the complex experimental approach tried to reflect that dynamic behavior of tumor cells in vitro and in vivo under drug treatment. The results are understandable and well presented. The figures are clear und the results reflect the figures. The cited literature is up to date and mainly from the last 5 years.

Major criticism

The presented data are well done. However, it is not clear which genetic background the different cell populations have. Since modern treatment of NSCLC are based on large scale next generation sequencing data to discern if an individual tumor population qualifies for additional drugs (such as EGFR, HER-2, ALK ROS-1 etc. treatment) it will be important to know this background in this experimental setting. Furthermore, I strongly recommend future experiments to use large scale RNA and protein profiling to come from a mechanistic approach to a more realistic biological approach that tries to reflect the tumor biology und the actual treatment circumstances.

Author Response

Reviewer #2 Comments and Suggestions for Authors

The author describe a model of cultured lung cancer cells with different genetic background and analyzed their behavior and interactions under the exposition with chemotherapeutic drugs. They used mathematical modelling to predict the cellular behavior under these conditions.

They concluded that coculture of sensitive and tolerant cells revealed that there is a dynamic competition and collaboration among these cell groups. Without cisplatin the sensitive cells dominate the tolerant cells but under cisplatin treatment the tolerant cells evade treatment with prolonged survival and proliferation whereas the sensitive cells undergo cell death. Further experiments were performed with tolerant and sensitive cells alone but with conditioned medium of one and another. This revealed an inhibitory effect on tolerant cells with conditioned medium of sensitive cells but not for the opposite experiment. Further experiments were performed to compare different drug exposure regimens and monitor sensitive and tolerant cells in an in vivo experiment with zebra fish larvae. Continous exposure of the drug showed a significant decrease in tolerant cells whereas intermittent drug exposure did not. They also tested the modulation of epigenetic changes by applying 5-AZA and SAHA showing that epigentic modulation is able to discern drug tolerance vs. true drug resistance.

The presented data are very interesting and the complex experimental approach tried to reflect that dynamic behavior of tumor cells in vitro and in vivo under drug treatment. The results are understandable and well presented. The figures are clear und the results reflect the figures. The cited literature is up to date and mainly from the last 5 years.

Major criticism

The presented data are well done. However, it is not clear which genetic background the different cell populations have. Since modern treatment of NSCLC are based on large scale next generation sequencing data to discern if an individual tumor population qualifies for additional drugs (such as EGFR, HER-2, ALK ROS-1 etc. treatment) it will be important to know this background in this experimental setting. Furthermore, I strongly recommend future experiments to use large scale RNA and protein profiling to come from a mechanistic approach to a more realistic biological approach that tries to reflect the tumor biology und the actual treatment circumstances.

Author’s response: We thank the reviewer for appreciating our work. We agree with the reviewer’s comments. Accordingly, we have noted the genetic alterations of each cell line using the cellosaurus database ,and  included them in the main text line number 105 - 107 . We also agree that large scale RNA and protein profiling data can help elucidate the mechanistic details. We definitely plan on doing these and hope to report them shortly.

Round 2

Reviewer 1 Report

The authors addressed all the points raised.